# The Effects of Salinity and Genotype on the Rhizospheric Mycobiomes in Date Palm Seedlings

**DOI:** 10.3390/biology13030190

**Published:** 2024-03-15

**Authors:** Mahmoud W. Yaish, Aya Al-Busaidi, Bernard R. Glick, Talaat Ahmed, Juha M. Alatalo

**Affiliations:** 1Department of Biology, College of Sciences, Sultan Qaboos University, P.O. Box 36, Muscat 123, Oman; albusaidiaya@gmail.com; 2Department of Biology, University of Waterloo, Waterloo, ON N2L 3G1, Canada; glick@sciborg.uwaterloo.ca; 3Environmental Science Center, Qatar University, Doha P.O. Box 2713, Qatar; t.alfattah@qu.edu.qa (T.A.); jalatalo@qu.edu.qa (J.M.A.)

**Keywords:** mycobiomes, date palm, PGPB, salinity, cultivars, NGS, rhizofungus

## Abstract

**Simple Summary:**

Salinity has a negative impact on crop production, yet this impact can be effectively mitigated using environmentally friendly plant growth-promoting microorganisms, which may naturally exist within the native microflora. Environmental conditions and the specific host plant control the presence and abundance of such microorganisms. As a first step toward the isolation of plant growth-promoting fungi, epiphytic mycobiomes associated with the salt-tolerant ‘Umsila’ and salt-susceptible ‘Zabad’ date palm cultivars when grown under saline and normal growth conditions were identified using internal transcribed spacer (ITS) rRNA sequences and next-generation sequencing (NGS) technology. A bioinformatic analysis revealed an apparent effect of salinity and the plant’s genotype on the fungal community structure. However, plant genotypes had no significant impact when grown under normal conditions. While the operational taxonomic units (OTUs) were annotated to *Acremonium*, *Acrocalymma*, *Agaricus*, *Aspergillus*, *Clonostachys*, *Fusarium*, *Penicillium*, and *Remersonia*, only a limited number of OTUs from the *Fusarium* and *Aspergillus* genera exhibit significant differential accumulation in response to salinity stress in date palms. The identified OTUs hold potential for advancing sustainable agricultural practices, as they could be further investigated as promising candidates for developing biofertilizers.

**Abstract:**

Salinity severely affects the health and productivity of plants, with root-associated microbes, including fungi, potentially playing a crucial role in mitigating this effect and promoting plant health. This study employed metagenomics to investigate differences in the structures of the epiphyte mycobiomes in the rhizospheres of seedlings of two distinct date palm cultivars with contrasting salinity tolerances, the susceptible cultivar, ‘Zabad’, and the tolerant cultivar, ‘Umsila’. Next-generation sequencing (NGS) of the internal transcribed spacer (ITS) rRNA was utilized as a DNA barcoding tool. The sequencing of 12 mycobiome libraries yielded 905,198 raw sequences of 268,829 high-quality reads that coded for 135 unique and annotatable operational taxonomic units (OTUs). An OTU analysis revealed differences in the rhizofungal community structures between the treatments regardless of genotype, and non-metric dimensional scaling (N-MDS) analyses demonstrated distinct separations between the cultivars under saline stress. However, these differences were not detected under the control environmental conditions, i.e., no salinity. The rhizospheric fungal community included four phyla (Ascomycota, Basidiomycota, Chytridiomycota, and Mucoromycota), with differences in the abundances of *Aspergillus*, *Clonostachys*, and *Fusarium* genera in response to salinity, regardless of the genotype. Differential pairwise comparisons showed that *Fusarium falciforme-solani* and *Aspergillus sydowii-versicolor* increased in abundance under saline conditions, providing potential future in vitro isolation guidelines for plant growth-promoting fungi. This study highlights the intricate dynamics of the rhizosphere microbial communities in date palms and their responses to salt stress. Additionally, we found no support for the hypothesis that indigenous epiphytic fungal communities are significantly involved in salinity tolerance in date palms.

## 1. Introduction

Salinity is a significant problem in plant agriculture, and problems caused by high salinity have become more severe in recent years, especially in arid and semiarid regions, due to both low precipitation rates and the overconsumption of groundwater for irrigation purposes [1,2]. Supporting plants with the required nutrition may enhance salinity tolerance [3]; however, using inorganic fertilizers may lead to environmental pollution, inappropriate changes in microbial community structure, and increased crop production costs [4]. 

Microbiota are composed of various microorganisms, including bacteria, fungi, archaea and cyanobacteria, that reside in a specific environment or host organism and are influenced by changes in their surroundings and reciprocally influence their environment [5]. Plant-associated fungi significantly impact plants’ growth and overall health [6]. They contribute to the growth and development of date palms (*Phoenix dactylifera*) [7,8] by producing growth-promoting substances, like auxins and cytokinins, which stimulate root development and enhance plant growth through increased cell division, elongation, and differentiation, thereby improving plant biomass production and vigor [9]. Moreover, certain fungi associated with date palm roots act as biocontrol agents, protecting the trees from harmful phytopathogens [10,11]. Fungi associated with date palm roots have garnered significant attention in scientific research due to their pivotal role in augmenting nutrient uptake processes [12] and improving stress tolerance, enabling date palms to thrive in challenging environmental conditions. Additionally, these fungi contribute to the plant’s resistance against drought, salinity [13], and heavy metal toxicity [14], by enhancing water and nutrient absorption and facilitating the accumulation of osmolytes and antioxidant compounds [7]. Fungi constitute integral elements within the microbiota associated with plants, assuming critical functions in the modulation of plant health, resilience to environmental stressors, dynamics of plant communities, and, consequently, the overall functioning of ecosystems [15,16].

The significance of mycorrhizae, fungi that form a symbiotic relationship with the roots of most plants, becomes more pronounced in the context of global warming and subsequent predicted increases in drought spells, especially in arid soils with high salt concentrations and extreme heat [17]. In such situations, root-associated fungi hold promise as cost-effective, environmentally friendly, and sustainable biofertilizers [18].

As a plant adapted to dry and saline environments, date palm is an excellent model tree for studying abiotic stress tolerance, and it holds considerable potential for mitigating deforestation in tropical and semi-tropical locales [19]. The endophytic microbiota associated with date palms has been previously investigated to elucidate its role in salt tolerance, resulting in the identification of diverse bacterial and fungal species that respond to abiotic stresses [20,21]. Recent research has demonstrated distinct variations in the epiphyte bacterial community structures between two date palm genotypes with different salinity tolerances under optimal growth conditions [22]. However, these differences tend to decrease when the plants are exposed to salinity stress, suggesting the existence of common salt-tolerant bacterial species capable of colonizing the roots of various date palm genotypes [22].

Traditional culture-based methods have limitations in effectively identifying fungi. However, culture-free approaches and next-generation sequencing (NGS) technologies provide versatile opportunities to comprehensively characterize microbial communities in the rhizosphere and understand the complex interactions between plants and microorganisms in the root zone under various environmental conditions [23].

In this report, we hypothesize that specific epiphytic fungi play an essential role in date palms acquiring salinity tolerance; therefore, this study examined the epiphytal fungal communities from the rhizospheres of two distinct date palm genotypes that exhibit contrasting responses to salinity stress. The primary objectives were to evaluate the influence of the date palm genotype on fungal community structure and investigate the potential involvement of these microorganisms in the salinity tolerance mechanism of date palms. The results reveal the presence of common OTU profiles in the rhizosphere among the cultivars of date palms when grown under control conditions. However, the fungal communities were slightly different in the rhizospheres of the two date palm cultivars when grown under saline conditions.

## 2. Materials and Methods

### 2.1. Soil Sampling, Seed Sterilization, and Plant Growth

The date palm seeds utilized in this study belong to the salt-tolerant ‘Umsila’ and salt-susceptible ‘Zabad’ cultivars, which resulted from the cross-pollination of a unique pollen genotype and were acquired from the experimental agriculture station of Sultan Qaboos University in Muscat, Oman. Prior to planting, the seeds underwent a series of treatments to ensure their sterility and optimize their germination, following a previously described protocol [22]. Briefly, the seeds were soaked in sterile distilled water overnight to remove any remaining fruit tissues. They underwent surface sterilization with 75% ethanol, followed by a 0.5% hypochlorite solution (liquid bleach) for five minutes per treatment. The seeds were subsequently washed with sterile water containing 10% Tween-20 and then rinsed five times with sterile distilled water. Afterward, the seeds were soaked in distilled water for 24 h to enhance germination and then planted in sterilized, moist vermiculite for ten days. Once the seeds germinated, they were transplanted into 4 L pots containing soil samples collected from various locations within the rhizospheres (0–20 cm depth) of date palm trees grown in the Al-Khoud area, Muscat, Sultanate of Oman (23°38′32.1″ N 58°11′42.8″ E).

The experimental design comprised eight pots allocated to each cultivar, with four pots designated for the control group (irrigated solely with water) and four for the treated group (irrigated with NaCl solutions). Each pot accommodated a minimum of four seedlings. While the control pot group was irrigated with sterilized distilled water as needed, the NaCl-treated pot group was irrigated with 50 mM saline water during the first week, with the NaCl concentration being gradually increased by 50 mM each subsequent week for the first month until reaching a concentration of 250 mM, which was then consistently used for watering the treated pots for the following eight weeks. At 12 weeks after planting, the roots were carefully separated from the pots, gently shaken to remove the larger soil particles, and then brushed to collect the proximal soil adhering to the roots. The plant pots were placed in a growth room under controlled environmental conditions during the experiment. The light was adjusted to a 16/8 h light/dark cycle, with a light intensity of 350 μE m^−2^ s^−1^ and maintaining day/night temperatures of 35/30 °C, respectively.

### 2.2. DNA Extraction and Illumina MiSeq Sequencing of ITS

Soil samples were processed to extract eDNA utilizing the TMPureLinkTM Genomic DNA Mini Kit (Invitrogen, Waltham, MA, USA), following the instructions specified by the manufacturer. The concentration and purity of the DNA samples were assessed using 1% TAE agarose gel electrophoresis and a Nanodrop2000 spectrophotometer (Thermo Scientific™, Waltham, MA, USA). The nuclear ribosomal internal transcribed spacer (ITS) region was utilized to barcode the fungal communities in the rhizospheres of the different date palm genotypes that grew under the control and salinity conditions.

ZymoBiomics Corporation, Irvine, CA, USA, conducted DNA sequencing as an outsourced service provider. DNA samples were employed for fungal community identification through a high-throughput sequencing of the ITS gene sequence using the Illumina^®^ MiSeq™ platform with a v3 reagent kit (600 cycles). For the targeted sequencing of the fungal ITS gene, the Quick-16S™ NGS Library Prep Kit was utilized, with custom ITS2 primers replacing the 16S primers TS-White90_ITS3F (5′-GCATCGATGAAGAACGCAGC-3′) and ITS_White90-ITS4R (5′-TCCTCCGCTTATTGATATGC-3′). A novel library preparation process was employed to ensure accurate results and minimize PCR chimera formation, utilizing real-time PCR machines to control the amplification cycles. The resulting amplicons were measured using quantitative PCR fluorescence readings and combined into pooled libraries with equal molarity that then underwent purification using the Select-a-Size DNA Clean & Concentrator™ from Zymo Research (Irvine, CA, USA). Subsequently, each library’s quantification was determined using the TapeStation^®^ system from Agilent Technologies (Santa Clara, CA, USA) and Qubit^®^ technology from Thermo Fisher Scientific (Waltham, MA, USA). Finally, each library was sequenced on an Illumina^®^ MiSeq™ platform with a v3 reagent kit (600 cycles), incorporating >10% PhiX spike-in during sequencing.

### 2.3. Bioinformatic Analysis

The unique sequences of DNA fragments (amplicons) were determined from the original data using the DADA2 pipeline, as described by Callahan et al. [24]. Any chimeric sequences were eliminated using the DADA2 pipeline. Taxonomic classification was carried out using the Uclust algorithm implemented in Qiime v.1.9.1, utilizing the Zymo Research Database as a curated reference for ITS sequences. Accordingly, OTUs were assigned. If applicable, the LEfSe algorithm [25] was employed with default parameters to identify taxonomic groups exhibiting significant abundance differences across different sample groups. To assess the diversity of the fungal communities concerning the control and salinity treatments, along with cultivar type, an ordination analysis was conducted using the Non-Metric Dimensional Scaling (N-MDS) method. The Bray–Curtis similarity index, utilized within the Past 3.0 software package, was employed for this analysis to quantify the similarity between samples [26]. A phylogenetic tree was also produced to illustrate the relationship of the fungal OTU ITS DNA sequences with abundance values reaching or surpassing 500 counts across diverse treatments.

A heatmap was generated to depict the fungal abundance profiles of OTUs exhibiting abundance values equal to or exceeding 500 counts across various treatments. The PermutMatrix software, version 1.9.3 [27], utilized default settings. Instances where OTUs displayed zero abundance values were substituted with half of the minimum count number. The Log_10_ transformation was also applied to the abundance values for analytical purposes.

Fungal identification data were subjected to pairwise comparisons to determine significant differences between groups using Tukey’s Honest Significant Difference (HSD) test as a post hoc analysis. Statistical analyses were performed using SPSS version 21.0 [28], with a significance level set at *p* ≤ 0.05.

## 3. Results

### 3.1. Metagenomic Analysis Revealed Minor Variations in Fungal Community Structures across Distinct Cultivars Due to Environmental Changes

The NGS sequencing of the 12 ITS libraries generated 905,198 raw sequences. Following trimming and filtration based on sequence size, 884,476 sequences were identified, and a total of 268,829 high-quality reads were obtained from the 12 libraries, annotated, and assigned to 670 operational taxonomic units (OTUs) (Appendix A).

Subsequently, the reads underwent rarefaction by excluding all unidentified OTUs, resulting in 331 annotated OTUs from the 12 libraries and 135 high-quality, unique reads clustered into OTUs. The sequences exhibited an average length of approximately 322 bp (Appendix A).

To determine the effect of salinity and cultivar type on the abundance of the OTUs, a comparative statistical analysis was performed on the resulting sequences of the ITS of the different samples (Appendix A). The analysis of variance as a factorial experiment showed that salt treatment significantly affected the OTU enrichment (*p* ≤ 0.0340). However, the results show that salinity treatment and the cultivar type did not influence the total number of reads or unique OTUs obtained when samples were compared separately (Table 1). There were slight differences in the number of OTU reads obtained from the samples; however, these differences were not statistically significant (*p* ≤ *0.05*). Furthermore, no significant differences were noted between the number of total OTUs obtained from the different samples.

### 3.2. Multidimensional Analysis Reveals Salinity Impact on Date Palm Rhizofungal Communities

To understand and identify a pattern of relationships between the cultivar type and the growth conditions of the date palm seedlings, N-MDS, as a multivariant analysis tool, was used to assess the distinctions between and among the 135 OTUs identified from the rhizosphere fungal communities in both cultivar types of date palm. The results show that salinity affected both cultivar types’ epiphytic fungal community structures. However, when subjected to control conditions, the results did not reveal different rhizofungal community structures between the date palm cultivars. The dissimilarities in the epiphytic rhizofungal community structures were more evident between cultivars under salinity stress, in which the analysis demonstrated a distinct separation of fungal communities in the three replicates (Figure 1).

### 3.3. Comparative Heatmap Analysis Reveals Treatment-Specific Fungal Abundance Patterns

A heatmap was constructed to facilitate the extraction of meaningful insights from the microbial count abundance profiles of the different treatments, where the OTUs and the samples were hierarchically clustered. The results reveal two sample clusters in which the control and salinity-treated samples were separated regardless of the associated cultivars (Figure 2). In most cases, the OTU count’s abundance clustering profile based on each fungal identity did not show any inter-replicate consistency. However, some OTUs showed some consistency and unique patterns among the treatments. For example, *Clonostachys rosea* and *Acrocalymma fici_1* were highly abundant in the control samples, while *Aspergillus tubingensis, Parathielavia appendiculata_1,* and *Remersonia tenuis* were abundant in the salinity-treated samples.

### 3.4. Epiphytic Fungal Diversity in Response to Plant Genotype and Salinity

The percentages of different taxa within each library were calculated to obtain insight into the compositions of the fungal communities of the two cultivars grown in environmental conditions. The date palm rhizospheric microbial community comprises Ascomycota, Basidiomycota, Chytridiomycota, and Mucoromycota phyla. The OTUs were classified into 14 classes, 25 orders, 45 families, 72 genera, and 124 species, genetic variants, or subspecies. At the phylum level, the results show an inconsistent distribution of the OTU abundance among the samples (Figure 3A). At the genus level, the results indicate that with both cultivars, *Aspergillus* spp. were more abundant in the salinity than in the control treatment, while *Clonostachys* spp. were more abundant under the control conditions. Additionally, the abundance of *Fusarium* spp. appears to be reduced in the salinity treatment in the ‘Zabad’ and, to some extent, in the ‘Umsila’ cultivar (Figure 3B).

Each cultivar’s rhizospheric fungal community structure was statistically compared to identify the differentially accumulated OTUs upon salinity treatment (Appendix A). A pairwise comparison between the fungal community structures of ‘Umsila’ and ‘Zabad’ grown under control and saline conditions revealed the presence of only two OTUs differentially accumulated at a significant (*p* ≤ 0.05) level in response to salinity and the plant’s genotype. *Fusarium falciforme-solani* exhibited a significant (*p* ≤ 0.05) increase in abundance when the ‘*Zabad*’ cultivar was grown under saline conditions, compared with the OTUs identified from the same cultivar grown under control conditions. Similarly, the abundance of *Aspergillus sydowii-versicolor* demonstrated a significant (*p* ≤ *0.05*) increase due to the salinity treatment of the ‘*Zabad*’ cultivar, and it was also more abundant in the ‘*Zabad*’ compared to the ‘*Umsilla*’ cultivar when subjected to salinity.

## 4. Discussion

Advanced DNA sequencing technologies, such as NGS, have the potential to enhance the efficient identification and subsequent successful isolation of microbial organisms [29]. Comparative microbiome analysis is a versatile means of identifying microbes that grow under conditions of high salinity in the rhizosphere of a specific plant genotype [22]. Therefore, this study employed NGS technology to study the mycobiomes of two distinct genotypes of date palms, each with different salinity tolerance capacities, to characterize mycobiomes in date palms.

A previous study on the rhizobacterial communities of the same date palm cultivars (‘Umsila’ and ‘Zabad’) found that the plant genotype did not significantly impact the epiphyte bacterial community structure under salinity stress [22]. In contrast, the results reported here reveal that the fungal communities are affected by salinity in the mycobiomes of both date palm genotypes. However, these fungal communities were similar under control conditions, indicating that the plant genotype did not play a role in determining the fungal community structure in the date palms when the two cultivars grew under control conditions. On the other hand, the results reported here show that salinity and cultivar type influenced the mycobiome structure in date palms. The varied abundances of OTUs can be attributed to the impact of salinity on distinct plant cultivars and the ensuing root physiological mechanisms linked to salinity tolerance, such as the biosynthesis and release of exudate metabolites that potentially enhance salinity tolerance and facilitate the recruitment of specific epiphytic fungal OTUs to the rhizosphere [30].

The *F. falciforme-solani* abundance was higher in the ‘*Zabad*’ cultivar when grown under saline conditions. This fungal OTU is the causative agent of root and stem rot disease in various plants [31,32,33]. Although salinity-induced plant vulnerability may allow opportunistic pathogens to proliferate, previous research has indicated that *F. solani* can exhibit beneficial effects under specific conditions. For instance, studies conducted in Ghana have demonstrated that *F. solani* can enhance plant resilience to biotic stress by synthesizing antibacterial secondary metabolites in *Chlorophora regia* [34]. Additionally, *F. solani* has been shown to alleviate abiotic stress by mitigating water stress in tomatoes [35]. An *A. sydowii* strain was differentially accumulated in ‘Zabad’ in response to salinity. Recently, *Aspergillus* species have been identified in marine settings, showing their capacity to thrive in environments with elevated salinity levels [36]. In this regard, *A. sydowii* is a halophilic fungus commonly found in different marine ecosystems and is considered a model organism for studying molecular adaptations of filamentous fungi to hyperosmolarity [37]. Recent studies showed that *A. sydowii* enhances plant growth by stimulating phosphorus solubilization in maize [38] and promoting higher phosphorus absorption by cotton plants [39]. The marine *A. versicolor* is a siderophore producer [40]. Given this information, both *F. solani* and *A. sydowii* have potential growth-promoting activities in date palm; however, this conjecture requires additional experimental verification. 

The high abundance of *A. sydowii-versicolor* in ‘Zabad’ mycobiomes relative to that in ‘Umsila’ mycobiomes under salinity stress may stem from the compatibility between this microorganism and the particular date palm cultivar rather than solely from the salt susceptibility of the cultivar However, an additional investigation is warranted to substantiate this hypothesis. Given that the ‘Umsila’ cultivar is genetically more salt-tolerant than the ‘Zabad’ cultivar, it is unlikely that the rhizospheric fungi described here influenced the salt tolerance levels in the ‘Umsila’ cultivar.

Prior investigations have reported a modest number of differentially accumulated OTUs within endophytic fungal communities isolated from the roots of date palm trees (cultivar: Khalas), wherein *Aspergillus niger* and *Humicola* sp. were the sole OTUs exhibiting significant accumulation under both saline and control conditions, respectively [21]. Likewise, the findings presented in this study demonstrate a limited set of differentially accumulated OTUs displaying significant alterations in response to salinity across both investigated cultivars, ‘Umsila’ and ‘Zabad’.

Our previously published results indicated that 16S rRNA sequencing provides a superior resolution compared to ITS sequencing when examining the bacterial community structure in date palm cultivars exposed to salinity conditions [21,22]. Several contributing factors may underlie this observation. It is possible that, under salinity stress, the bacterial community may exert dominance over the fungal community, especially if the bacteria assume a more significant role in the rhizosphere under salinity. Nonetheless, the complex relationship between the bacterial and fungal community structures necessitates further investigation in the context of date palms.

A previous study om soil microbial communities indicated that deterministic processes played a role in shaping the structure of a bacterial community. In contrast, the fungal community exhibited a more significant influence of stochastic processes in the primary succession [41], and the accuracy of the ITS data is highly dependent on the metabarcoding pipeline [42]. While the ITS region was previously used for microbiome fungal identification in date palm [4,21,43], it may have limitations where some fungal groups have conserved ITS sequences, making it challenging to distinguish between closely related species, and therefore, the community structure information obtained based on ITS sequences could be misleading [44]. The response of microbial communities to salinity is usually complex [45]. Nevertheless, if the bacterial community structure undergoes more significant changes or adaptations under salinity conditions, the 16S rRNA marker might better capture these dynamics.

Previous research showed that irrigation with saline groundwater reduces date palm root-associated fungal abundance and amends their community structural prototypes [46]. Fungi use sporulation as an adaptation mechanism to deal with salinity [41,47]. Utilizing a standard DNA extraction protocol in the DNA barcoding method to identify fungal communities may yield DNA predominantly from non-sporulated fungi, making the presence of fungal spores from various OTUs likely undetectable and subsequently underrepresented within the fungal communities. Recent findings revealed an inverse correlation between soil salinity and mycorrhizal colonization, along with a positive correlation between easily extractable glomalin-related soil protein (an indicator of symbiotic relationships with plant roots) and spore density in date palm oases across varying salinity levels in the Tunisian Sahara [48]. However, the current study did not investigate the presence of fungal spores.

## 5. Conclusions

The composition of the plant-associated mycobiome can be influenced by both the host plant and salinity levels, leading to alterations in the mycobiome’s structure. The rhizospheric mycobiome identified in this study is affected by salinity in date palms but did not show significant differences when the plants were grown under control conditions. While they share similar OTU abundance profiles, the genotype of the plant plays a role in determining the structure of the mycobiome when the plant grows in the presence of salinity. A few OTUs of the *Fusarium* and *Aspergillus* genera are differentially accumulated to a significant extent due to salinity in date palms. The identification and subsequent isolation of plant-associated growth-promoting microbes, including fungi, can potentially contribute to developing genotype-specific biofertilizers to mitigate salinity in the future.

## Figures and Tables

**Figure 1 biology-13-00190-f001:**
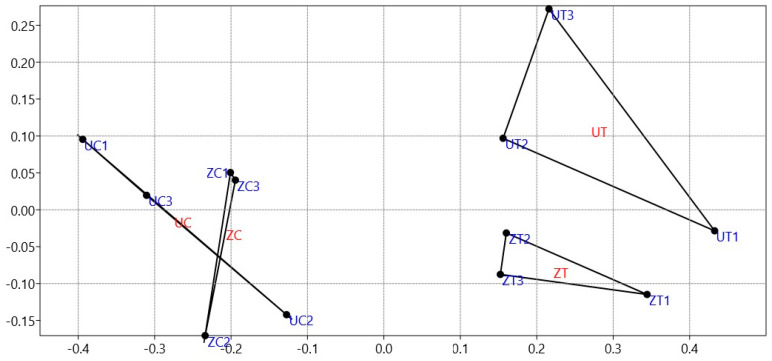
Non-metric dimensional scaling (N-MDS) utilizing a Bray–Curtis matrix illustrates the distances among the 135 identified OTUs from the rhizospheric fungal communities of ‘Umsila’ (UC) and ‘Zabad’ (ZC) cultivars under control conditions and ‘Umsila’ (UT) and ‘Zabad’ (ZT) cultivars under salinity stress.

**Figure 2 biology-13-00190-f002:**
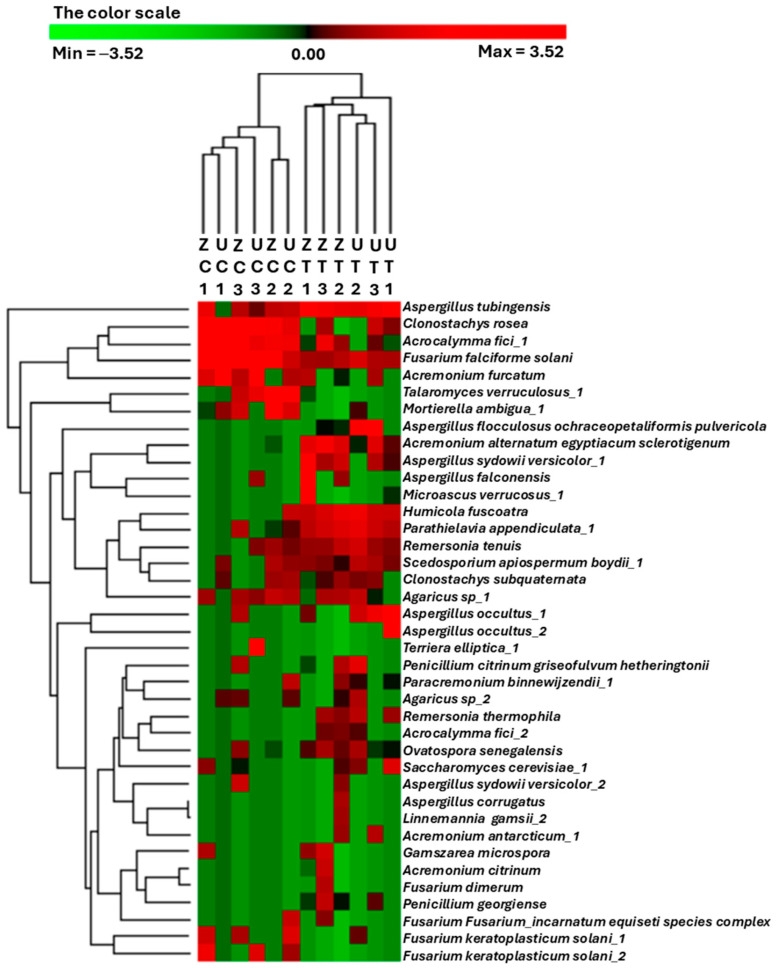
A heatmap generated from a hierarchical cluster analysis and a dendrogram depicting the normalized relative abundance of the top 39 most abundant operational taxonomic units (OTUs) identified from the rhizospheric fungal communities of ‘Umsila’ (UC) and ‘Zabad’ (ZC) cultivars under control conditions, as well as ‘Umsila’ (UT) and ‘Zabad’ (ZT) cultivars under saline conditions.

**Figure 3 biology-13-00190-f003:**
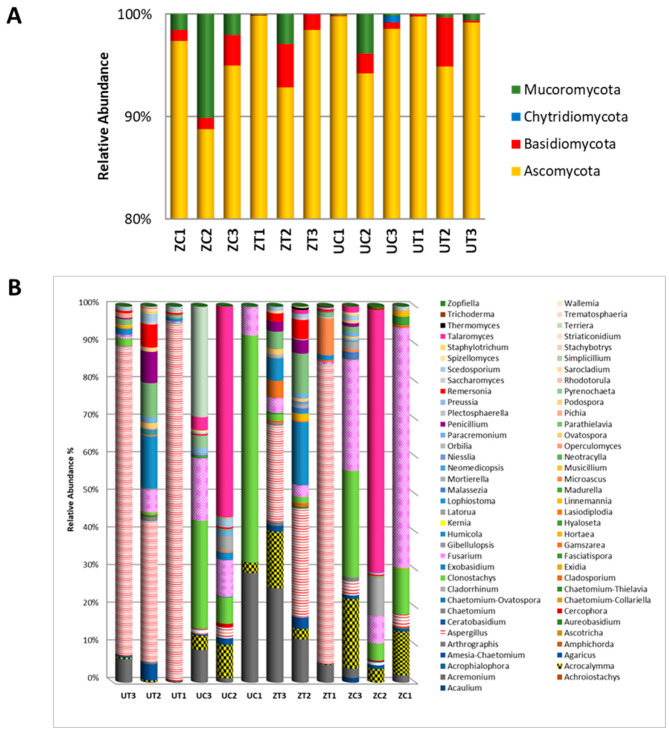
The OTUs within the fungal community are depicted in (**A**) at the phylum level and (**B**) at the genus level for the ‘Umsila’ (UC) and ‘Zabad’ (ZC) cultivars under control growth conditions, as well as for the ‘Umsila’ (UT) and ‘Zabad’ (ZT) cultivars under saline conditions. The abundance is shown as a percentage relative to the total number of reads per OTU.

**Table 1 biology-13-00190-t001:** The count of reads and unique OTUs obtained from the ITS sequencing of each sample replicate (R) from the rhizospheric fungal communities of ‘Umsila’ (UC) and ‘Zabad’ (ZC) cultivars under control conditions, as well as ‘Umsila’ (UT) and ‘Zabad’ (ZT) cultivars under salinity stress, were determined.

Number of Reads						
	R1	R2	R3	Average	Comparison	*p*-Value
ZC	11,297	18,693	17,903	15,964	ZC vs. ZT	0.1338
ZT	25,098	28,667	36,091	29,952	UC vs. UT	1.0
UC	27,727	23,607	13,800	21,711	ZT vs. UT	0.5187
UT	31,704	21,278	12,964	21,982	ZC vs. UC	0.7385
**OTU Enrichment**						
	**R1**	**R2**	**R3**	**Average**	**Pairwise Comparison**	***p*-Value**
ZC	20	21	34	25	ZC vs. ZT	0.37
ZT	33	42	31	35	UC vs. UT	0.30
UC	9	25	24	19	ZT vs. UT	0.86
UT	27	38	27	31	ZC vs. UC	0.78

## Data Availability

The sequence datasets generated and/or analyzed during the current study are available from the NCBI repository: http://www.ncbi.nlm.nih.gov/bioproject/1071109 (13 March 2024).

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
