# Peer review of "The Effects of Salinity and Genotype on the Rhizospheric Mycobiomes in Date Palm Seedlings"

_biology, 2024, doi:10.3390/biology13030190_

Round 1

Reviewer 1 Report

Comments and Suggestions for Authors

The objective of this study is to examine the impact of salinity on the fungal population within the rhizospheric soil of two cultivars of date palm seedlings : Umsila (salt-tolerant) and Zabad (salt-susceptible). The results indicate that both cultivars share a common fungal community under control conditions. However, the addition of salt induces changes in the presence and abundance of certain fungal OTUs. Clonostachys rosea and Acrocalymma fici_1 were highly abundant in the control samples, while Aspergillus tubingensis, Parathielavia appendiculata_1, and Remersonia tenuis were abundant in the salinity-treated samples. Fusairum OTUs were more abundant in Zabad under control treatment.  These results underscore the importance of rhizospheric fungi in plants' ability to withstand salt stress. They also help identify fungal species that could serve as promising candidates for biofertilizers on plants with low salt tolerance in arid and semi-arid regions, considering the plant-fungus compatibility. The article is very well-written, the writing is clear, and the reading is very enjoyable. The data presented are clear, although personally, I don't quite see the added value of Figure 4. Some restructuring and reformulation would strengthen the message and eliminate redundancy.

Title : I suggest to add « Date Palms seedlings »

Abstract

l. 24. This is not a good start for an abstract. It should begin with the context, the problem statement (hypothesis), and then the methodology.

Introduction

l. 30-32. It probably lacks a bit of precision. OTU analysis revealed differences in fungal communities between what and what? Also, NMDS indeed allow distinguishing fungal communities between cultivars, as well as between control/salt conditions (Fig 1).

l. 47. « are influenced by changes in their surrounding and reciprocally. » This makes the connection with the next paragraph, which lists the ways in which microorganisms can be useful for plants in relation to their environment.

In addition, a little restructuring would enable common properties to be grouped together, e.g. l. 54 "enhance nutrient uptake" and l. 57-57, "improving water and nutrient absorption".

l. 49. The introduction to the plant studied here is somewhat abrupt. Perhaps simply stating here in one sentence why it has been the subject of numerous studies regarding AMF would suffice.

l. 80-83. Since the design of a biofertilizer is not the focus of the paper, it does not seem relevant to include it in the introduction. This could confuse the message, as readers might expect the results to provide information regarding the development of « biofertilizer formulations ».

l. 91. « the influence of date palm/plants genotype »

l.96 – 97. It's unnecessary to mention what is not addressed in the paper.

Materials and methods

Specify here that Umsila is salt-tolerant and Zabad is salt-susceptible; this information only appears in the Simple Summary and abstract.

l.125. 350 µE m-2.s-1

l.156. (OTUs) (not « (OUTs)were »)

Results

All right, although I don't quite see the significance of Figure 4.

Discussion

l. 277-294. This paragraph should be in the introduction; these are not discussion points. And once again, mentioning biofertilizers seems a bit curious since the results do not provide details on this subject.

l.302-313. A bit difficult to understand. On one hand, it causes rotting of stems and roots, and on the other hand, it contributes to improving stress resistance? There needs to be a transition sentence. Also, the phrase "enhance stress" / « improve stress » isn't very intuitive; I understood it as "improve stress tolerance". If that's the case, it should be formulated as such.

l.325-327. Very interesting. References? (if you have it)

This point also constitutes an interesting element in relation to the concept of "biofertilizer", as the notion of fungi-plant compatibility is fundamental.

l.330-336. To facilitate understanding, I would start by discussing the "prior investigations" and then present the obtained results. The sentences are also a bit too long, making the reading complicated.

Conclusion

The concept of "biofertilizers" muddles the message and doesn't adequately highlight the results obtained here. The idea of designing biofertilizers can be retained, but perhaps presented as a perspective. In conclusion, it's essential to reinforce the idea of an interaction between "salt-mycobiome-cultivars," as demonstrated earlier (lines 302-306), as this is what the paper illustrates.

Author Response

Reviewer-1

Comment: The objective of this study is to examine the impact of salinity on the fungal population within the rhizospheric soil of two cultivars of date palm seedlings : Umsila (salt-tolerant) and Zabad (salt-susceptible). The results indicate that both cultivars share a common fungal community under control conditions. However, the addition of salt induces changes in the presence and abundance of certain fungal OTUs. Clonostachys rosea and Acrocalymma fici_1 were highly abundant in the control samples, while Aspergillus tubingensisParathielavia appendiculata_1, and Remersonia tenuis were abundant in the salinity-treated samples. Fusairum OTUs were more abundant in Zabad under control treatment.  These results underscore the importance of rhizospheric fungi in plants' ability to withstand salt stress. They also help identify fungal species that could serve as promising candidates for biofertilizers on plants with low salt tolerance in arid and semi-arid regions, considering the plant-fungus compatibility. The article is very well-written, the writing is clear, and the reading is very enjoyable. The data presented are clear, although personally, I don't quite see the added value of Figure 4. Some restructuring and reformulation would strengthen the message and eliminate redundancy.

Reply: Figure 4 and the related information was deleted from the manuscript as suggested by the reviewer.

Comment: Title : I suggest to add « Date Palms seedlings »

Reply: The word “seedling” word was added to the title as suggested by the reviewer.

Abstract

Comment: l. 24. This is not a good start for an abstract. It should begin with the context, the problem statement (hypothesis), and then the methodology.

Reply: As suggested by the reviewer, a sentence was added to introduce the problem statement for the context.

Introduction

Comment: l. 30-32. It probably lacks a bit of precision. OTU analysis revealed differences in fungal communities between what and what? Also, NMDS indeed allow distinguishing fungal communities between cultivars, as well as between control/salt conditions (Fig 1).

Reply: Clarified (in the abstract), as suggested by the reviewer.

Comment: l. 47. « are influenced by changes in their surrounding and reciprocally. » This makes the connection with the next paragraph, which lists the ways in which microorganisms can be useful for plants in relation to their environment.

Reply: Corrected as suggested by the reviewer.

Comment: In addition, a little restructuring would enable common properties to be grouped together, e.g. l. 54 "enhance nutrient uptake" and l. 57-57, "improving water and nutrient absorption".

  1. 49. The introduction to the plant studied here is somewhat abrupt. Perhaps simply stating here in one sentence why it has been the subject of numerous studies regarding AMF would suffice.

Reply: The paragraph was restructured as the reviewer suggested.

Comment: l. 80-83. Since the design of a biofertilizer is not the focus of the paper, it does not seem relevant to include it in the introduction. This could confuse the message, as readers might expect the results to provide information regarding the development of « biofertilizer formulations ».

Reply: The information regarding biofertilizers (l. 80-83) was deleted from the introduction section as suggested by the reviewer.

Comment: l. 91. « the influence of date palm/plants genotype »

Reply: Corrected as suggested by the reviewer. Thank you.

Comment: l.96 – 97. It's unnecessary to mention what is not addressed in the paper.

Reply: This information was deleted from the introduction, as suggested by the reviewer.

Materials and methods

Comment: Specify here that Umsila is salt-tolerant and Zabad is salt-susceptible; this information only appears in the Simple Summary and abstract.

Reply: The information was added as suggested by the reviewer.

Comment: l.125. 350 µE m-2.s-1

Reply: Corrected as suggested by the reviewer. Thank you.

Comment: l.156. (OTUs) (not « (OUTs)were »)

Reply: Corrected as suggested by the reviewer.

Results

Comment: All right, although I don't quite see the significance of Figure 4.

Reply: We agree with the reviewer, therefore, Figure 4 and the related information was deleted from the manuscript.

Discussion

Comment: l. 277-294. This paragraph should be in the introduction; these are not discussion points. And once again, mentioning biofertilizers seems a bit curious since the results do not provide details on this subject.

Reply: Part of the paragraph was moved to the introduction section, and the other parts that describe biofertilizers were deleted as suggested by the reviewer.

Comment: l.302-313. A bit difficult to understand. On one hand, it causes rotting of stems and roots, and on the other hand, it contributes to improving stress resistance? There needs to be a transition sentence. Also, the phrase "enhance stress" / « improve stress » isn't very intuitive; I understood it as "improve stress tolerance". If that's the case, it should be formulated as such.

Reply: The paragraph was restructured and corrected as the reviewer suggested.

Comment: l.325-327. Very interesting. References? (if you have it)

This point also constitutes an interesting element in relation to the concept of "biofertilizer", as the notion of fungi-plant compatibility is fundamental.

Reply: In fact, no reference is available therefore, the paragraph was restructured to have this piece of information as a hypothesis, which requires further investigation to prove.

Comment: l.330-336. To facilitate understanding, I would start by discussing the "prior investigations" and then present the obtained results. The sentences are also a bit too long, making the reading complicated.

Reply: The paragraph was rearranged, and some sentences were rewritten as suggested by the reviewer.

Comment: Conclusion

The concept of "biofertilizers" muddles the message and doesn't adequately highlight the results obtained here. The idea of designing biofertilizers can be retained, but perhaps presented as a perspective. In conclusion, it's essential to reinforce the idea of an interaction between "salt-mycobiome-cultivars," as demonstrated earlier (lines 302-306), as this is what the paper illustrates.

Reply: In response to the reviewer's feedback, we have revised the conclusion section accordingly. Furthermore, we have reduced the emphasis on "biofertilizer" related concepts in the manuscript to better align with the content.

Reviewer 2 Report

Comments and Suggestions for Authors

The manuscript is ok and easy to read.

In line 30, I indicated the word "some," which I have crossed out.

In my opinnion the conclusions section can be improve 

Author Response

Reviewer 2:

Comment: The manuscript is ok and easy to read.

In line 30, I indicated the word "some," which I have crossed out.

Reply: Corrected as suggested by the reviewer. Thank you.

Comment: In my opinnion the conclusions section can be improve 

Reply: The conclusion was improved, as suggested by the reviewer.

Reviewer 3 Report

Comments and Suggestions for Authors

-Lines 60-68: Fungi can resistant in some extreme events such as fire. Please see and include this paper [doi:10.1016/j.apsoil.2024.105303]. Moreover, soil bacteria is stable in undisturbed environment and sensitive to changing soil condition [doi: 10.3389/fmicb.2023.1285445].

- Lines 89-97: Please provide the hypothesis of the study.

-Line 98: Weather conditions of study area should be provided.

- How deep of soil samples were taken? How many replication?

- Figure 2: the text are bad resolution.

-Lines 283-286: Biofertilizers can improve the soil biodiversity. Please see this paper [doi.org/10.3390/agronomy11061116]

Comments on the Quality of English Language

-

Author Response

Reviewer 3:

Comment: -Lines 60-68: Fungi can resistant in some extreme events such as fire. Please see and include this paper [doi:10.1016/j.apsoil.2024.105303]. Moreover, soil bacteria is stable in undisturbed environment and sensitive to changing soil condition [doi: 10.3389/fmicb.2023.1285445].

Reply: The additional information recommended by the reviewer has been incorporated into the manuscript, including the citation of one of the suggested references. However, the remaining reference is deemed less pertinent to the subject matter under discussion in the manuscript. Thank you.

Comment: - Lines 89-97: Please provide the hypothesis of the study.

Reply: The information was added to the manuscript as suggested by the reviewer.

Comment: -Line 98: Weather conditions of study area should be provided.

Reply: The study was carried out under controlled environmental conditions as mentioned in the materials and method section of the manuscript.

Comment: - How deep of soil samples were taken? How many replication?

Reply: The information was added to the manuscript as the reviewer suggested.

- Figure 2: the text are bad resolution.

Reply: The original figure was generated by software and embedded within the text of the previous revision. In the new revision, we have improved the quality of the figure as suggested by the reviewer. Thank you.

Comment: -Lines 283-286: Biofertilizers can improve the soil biodiversity. Please see this paper [doi.org/10.3390/agronomy11061116]

Reply: Other reviewers have recommended removing most of the information about "biofertilizers" from the manuscript, as it is not aligned with the core concept of the current manuscript. Thank you very much. I apologize for not adding the information mentioned in the manuscript. Thank you.

Round 2

Reviewer 3 Report

Comments and Suggestions for Authors

Accept in present form

Comments on the Quality of English Language

-